

Earth System
Dynamics

# Population exposure to droughts in China under the 1.5 °C global warming target

**Jie Chen**[1,2], **Yujie Liu**[1], **Tao Pan**[1], **Yanhua Liu**[1], **Fubao Sun**[1], and **Quansheng Ge**[1]

[1]Institute of Geographic Sciences and Natural Resources Research,
Chinese Academy of Sciences (CAS), Beijing, 100101, PR China
[2]University of Chinese Academy of Sciences (UCAS), Beijing, 100049, PR China

**Correspondence:** Yujie Liu (liuyujie@igsnrr.ac.cn) and Tao Pan (pantao@igsnrr.ac.cn)

**Abstract.** The Paris Agreement proposes a 1.5 °C target to limit the increase in global mean temperature (GMT). Studying the population exposure to droughts under this 1.5 °C target will be helpful in guiding new policies that mitigate and adapt to disaster risks under climate change. Based on simulations from the Inter-Sectoral Impact Model Intercomparison Project (ISI-MIP), the Standardized Precipitation Evapotranspiration Index (SPEI) was used to calculate drought frequencies in the reference period (1986–2005) and 1.5 °C global warming scenario (2020–2039 in RCP2.6). Then population exposure was evaluated by combining drought frequency with simulated population data from shared socioeconomic pathways (SSPs). In addition, the relative importance of climate and demographic change and the cumulative probability of exposure change were analyzed. Results revealed that population exposure to droughts in the east of China is higher than that in the west; exposure in the middle and lower reaches of the Yangtze River region is the highest, and it is lowest in the Qinghai-Tibet region. An additional TSI 6.97 million people will be exposed to droughts under the 1.5 °C global warming scenario relative to the reference period. Demographic change is the primary contributor to exposure (79.95 %) in the 1.5 °C global warming scenario, more than climate change (29.93 %) or the interaction effect (−9.88 %). Of the three drought intensities – mild, moderate, and extreme – moderate droughts contribute the most to exposure (63.59 %). Probabilities of increasing or decreasing total drought frequency are roughly equal (49.86 % and 49.66 %, respectively), while the frequency of extreme drought is likely to decrease (71.83 % probability) in the 1.5 °C global warming scenario. The study suggested that reaching the 1.5 °C target is a potential way for mitigating the impact of climate change on both drought hazard and population exposure.

## 1 Introduction

The goal of the Paris Agreement is to pursue efforts to limit the increase in global mean temperature (GMT) to 1.5 °C above preindustrial levels, recognizing that this limit would significantly reduce the risks and impacts of climate change (UNFCCC, 2015). Studies quantifying climate extreme events and their socioeconomic impacts under the 1.5 °C target are urgently needed. These types of studies are key content for the IPCC special report on the 1.5 °C target, which will be published in 2018. Risk is often represented as the probability of occurrence of hazardous events or trends multiplied by the impacts if these events or trends occur, it results from the interaction of hazard, exposure, and vulnerability (Field et al., 2014). Therefore, exposure assessment is one of the most important aspect of disaster risk assessment. Exposure usually refers to the presence of people, livelihoods, species, or ecosystems; environmental functions, services, and resources; infrastructure; or economic, social, or cultural assets in places and settings that could be adversely affected (Field et al., 2014). As one of the most devastating natural disasters, droughts rank first in terms of globally affected populations (Mishra and Singh, 2010), and the frequency and intensity of droughts are likely to increase with global warming (Stocker et al., 2014; Field et al., 2012). De-

mographic growth in droughts-prone locations can increase the population exposed, and ultimately lead to increased risk (Forzieri et al., 2017; United Nations, 2013). Droughts have large impacts in China due to typical continental monsoon climate conditions and the large population (Qin et al., 2015). The losses caused by droughts accounted for 19.4 % of all meteorological disasters from 1985 to 2014 (CMA, 2015). Therefore, research on population exposure to droughts in China under the 1.5 °C target will be important for understanding future risk.

Several studies of the 1.5 °C target have been conducted recently (Donnelly et al., 2017; Henley and King, 2017; Huntingford et al., 2017; Guiot and Cramer, 2016). The objectives have been to evaluate the possible greenhouse gas (GHG) emission pathways to achieve the 1.5 °C target (CAT, 2016; Mitchell et al., 2017) or predict changes in extreme climate events under the 1.5 °C target (Karmalkar and Bradley, 2017; King et al., 2017; Wang et al., 2017). However, the influence of climate change on socioeconomic aspects, which also needs detailed assessment, has received less attention. The effects of droughts on human populations need to be quantified to identify the locations and intensity of disasters to which people will be exposed under the 1.5 °C target. Smirnov et al. (2016) assessed changing population exposure to extreme droughts in RCP8.5 using the Standardized Precipitation Evapotranspiration Index (SPEI); their results indicated that population exposure would increase 426.6 % compared to current conditions. RCP8.5 is a high emission scenario, which is far more than the 1.5 °C target, and the study did not account for mild and moderate droughts. Sun et al. (2017) analyzed population exposure to droughts under 1.5 and 2.0 °C global warming scenarios in the Haihe River Basin based on SPEI; their results indicated that population exposure under 1.5 °C conditions would be reduced 30.4 % relative to 1986–2005. However, population data used in this study was from the sixth national population census of China in 2010, in both reference period and global warming scenarios, ignoring the impact of demographic growth on population exposure change. In addition to climate change, the number, growth, and spatial distribution of population are important contributors to exposure risk, and should be taken into consideration.

In this study, population exposure to droughts under global warming was quantified, the relative importance of different factors, and the uncertainty in exposure change were evaluated. First, SPEI was used to calculate drought frequencies during the reference period and 1.5 °C global warming scenario based on simulations from the Inter-Sectoral Impact Model Intercomparison Project (ISI-MIP; Warszawski et al., 2014). Second, modeled population data from shared socioeconomic pathways (SSPs; O'Neill et al., 2014) were used to evaluate the spatial distribution and change in population exposure to droughts in China. Third, the relative importance of climate and demographic change was compared, and the uncertainty in exposure change was assessed using cumulative distribution functions (CDFs). This evaluation of population exposure to droughts in China under the 1.5 °C target is expected to provide a basis for adaptation and mitigation strategies.

## 2 Materials and methods

### 2.1 Materials

Meteorological data, including precipitation, average maximum temperature, average minimum temperature, average wind speed, average relative humidity, and solar radiation, used in this study were obtained from ISI-MIP (Warszawski et al., 2014), which contains five global climate models (GCMs) simulation results in representative concentration pathways (RCPs). These five GCMs are GFDL-ESM2M, HadGEM2-ES, IPSL-CM5A-LR, MIROC-ESM-CHEM, and NorESM1-M. In this study, we synthesized the results of the five GCMs based on the separately calculated SPEI for each GCM, as combining results of multiple models has been shown to be superior to a single model (Zhou and Yu, 2006). The chosen reference period was 1986–2005, which is a common period to assess climate change effect, and is 0.61 °C warmer than preindustrial levels (Stocker et al., 2014). According to previous research (Schleussner et al., 2016; Sun et al., 2017), a stable increase of 1.5 °C GMT above the preindustrial level for 20 years will be in 2020–2039 under RCP2.6. As for population data, the United Nations (http://www.un.org, last access: 20 August 2018), World Bank (https://data.worldbank.org.cn, last access: 20 August 2018), and other organizations have proposed projections of population in the future. Considering the RCP scenario we chose in this study, projections of population data in SSP scenarios were used. These are reference pathways describing plausible alternative trends in the evolution of society and ecosystems over a century timescale, in the absence of climate change or climate policies. According to the correspondence between RCPs and SSPs provided by the IPCC, RCP2.6 generally corresponds to SSP1. SSP1 is a sustainable development scenario facing low mitigation and adaption challenges (O'Neill et al., 2014). Therefore, SSP1 was chosen in this study. Population data for SSP1 was obtained from the National Institute for Environmental Studies (NIES), Japan, which was downscaled from the International Institute for Applied Systems Analysis (IIASA) simulated results. Populations in 2000 and 2030 were used to represent the populations in the reference period and 1.5 °C global warming scenario, respectively. The spatial resolution of meteorological and population data is given in degree latitude and longitude, i.e., $0.5° \times 0.5°$.

## 2.2 Calculation of SPEI

Combining the characteristics of the Standardized Precipitation Index (SPI; McKee et al., 1993) at multiple scales and the Palmer Drought Severity Index (PDSI; Palmer, 1965), which is sensitive to warming, SPEI was proposed by Vicente-Serrano et al. (2010). The SPEI reflects the change in water deficit using the log-logistic probability distribution function, and obtains the drought index value by standardizing. SPEI is commonly applied as an indication of meteorological droughts and, to a lesser extent, hydrological droughts (Zargar et al., 2011; Hao et al., 2018). SPEI-12 was chosen in this study to reflect long-term trends and interannual changes in droughts. Differences between precipitation ($P$) and potential evapotranspiration ($\mathrm{ET}_0$), which reflect the water surplus or deficit in a region, were calculated to deduce the SPEI by using

$$D = P - \mathrm{ET}_0. \tag{1}$$

The Thornthwaite (1948) equation for $\mathrm{ET}_0$ in SPEI only takes temperature into account, ignoring the effects other dynamic factors on droughts. Therefore, the Penman–Monteith equation (Allen et al., 1998) was replaced to calculate $\mathrm{ET}_0$ in this study. The Penman–Monteith equation comprehensively considers the impact of both thermal and dynamic factors on $\mathrm{ET}_0$, i.e., temperature, wind speed, relative humidity, and solar radiation. Therefore, results are more consistent with true reference crop evapotranspiration. The radiation coefficient used is based on the radiation calibration results in China provided by Yin et al. (2008):

$$\mathrm{ET}_0 = \frac{0.408 \Delta \left(R_{\mathrm{n}} - G\right) + \gamma \frac{900}{T+273} u_2 (e_{\mathrm{s}} - e_{\mathrm{a}})}{\Delta + \gamma (1 + 0.34) u_2}. \tag{2}$$

Here, $\mathrm{ET}_0$ is the potential evapotranspiration; $R_{\mathrm{n}}$ is the net radiation; $G$ is the soil heat flux density; $T$ is the surface mean daily air temperature; $u_2$ is the wind speed at 2 m height above the ground; $e_{\mathrm{s}}$ is the saturation vapor pressure; and $e_{\mathrm{a}}$ is the actual vapor pressure. The SPEI was calculated by the R SPEI package (https://CRAN.R-project.org/package=SPEI, last access: 20 August 2018). The input data are monthly time series of $D$ (differences between precipitation and potential evapotranspiration), where the set parameters are scale = 12, kernel = "rectangular", distribution = "log-logistic", and fit = "ub-pwm". The categorization of drought grade by SPEI and its probability as well as the definition of each grade of drought are shown in Table 1 (Liu and Jiang, 2015).

## 2.3 Population exposure to drought

Our measure of population exposure is the number of people exposed to mild, moderate, and extreme droughts. That is, the annual average percentage of mild, moderate, and extreme droughts multiplied by the number of people exposed

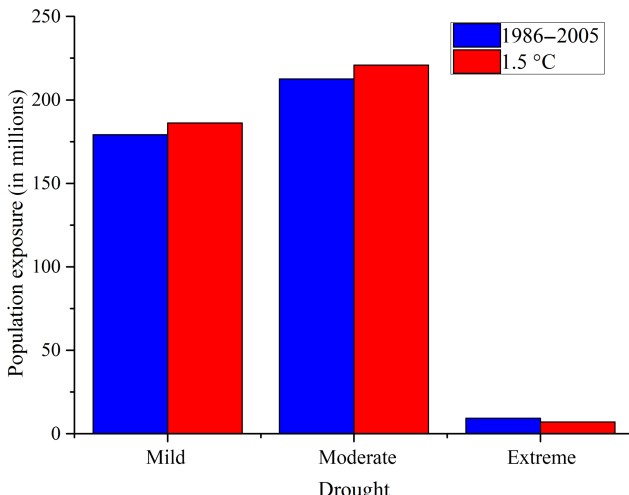

**Figure 1.** Population exposure to mild, moderate, and extreme droughts for the reference period (1986–2005) and 1.5 °C global warming scenario (2020–2039 in RCP2.6).

to that outcome (Jones et al., 2015). In this study, population exposures to mild, moderate, and extreme droughts were calculated in the 1.5 °C global warming scenario (2020–2039 in RCP2.6) and compared to the results of the reference period (1986–2005). The spatial distribution and change in exposure were analyzed based on the regional separation of China's population into eight major demographic regions (Hu, 1990; Fig. S1 in the Supplement).

## 2.4 Relative importance and cumulative probability analysis

Population exposure change was decomposed into climate change, demographic change, and interaction effects to evaluate the relative importance using the techniques from a previous study (Jones et al., 2015). The impact of population was calculated by holding climate constant; that is, the frequency of mild, moderate, and extreme droughts in the reference period multiplied by the population in the SSP1 scenario. Similarly, when calculating the impact of climate, the population was held constant; that is, the frequency of mild, moderate, and extreme droughts in the RCP2.6 scenario was multiplied by the population in the reference period. The interaction effect was also evaluated to assess whether the area with continued population growth is experiencing more drought events under climate change.

The CDF of a random variable $X$ is the function representing the probability that the random variable $X$ takes on a value less than or equal to $x$. The uncertainty in drought frequency and exposure change were analyzed based on CDFs to evaluate the possible impact of climate change. First, the change in frequency and population exposure to mild, moderate, extreme, and all droughts were separately calculated in the 1.5 °C global warming scenario relative to the refer-

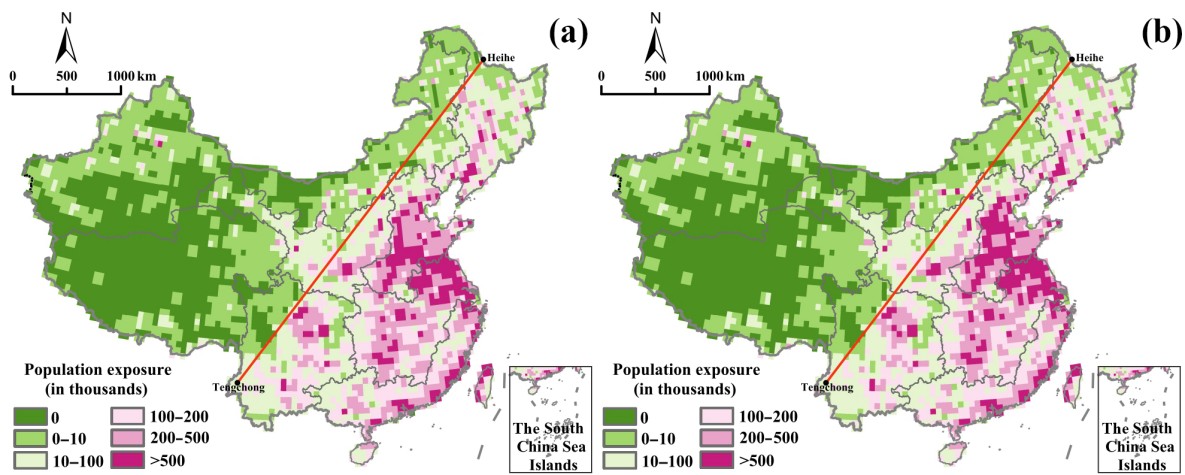

**Figure 2.** Spatial distribution of population exposure to droughts in (a) the reference period (1986–2005) and (b) the 1.5 °C global warming scenario (2020–2039 in RCP2.6). (Red line is the Hu line, an imaginary line that diagonally divides the area of China into two parts, stretching from the city of Heihe in Heilongjiang Province to Tengchong in Yunnan Province, which is also called the "geo-demographic demarcation line".) Please note that the above figure contains disputed territories.

**Figure 3.** Change in population exposure to droughts between the reference period (1986–2005) and the 1.5 °C global warming scenario (2020–2039 in RCP2.6), (a) change in exposure to mild droughts, (b) change in exposure to moderate droughts, (c) change in exposure to extreme droughts, and (d) change in exposure to all droughts. Please note that the above figure contains disputed territories.

**Table 1.** Drought grade categories and probability in the SPEI and its definition.

| SPEI | Categories | Probability | Definition |
|---|---|---|---|
| $> -0.5$ | Normal and wetness | 69.15 % | Precipitation is normal or more than normal, surface is wet, and there is no drought |
| $-1.0$ to $-0.5$ | Mild drought | 14.98 % | Precipitation is less than normal, surface air is dry, and soil moisture is insufficient |
| $-2.0$ to $-1.0$ | Moderate drought | 13.59% | Precipitation continued to be less than normal, surface is dry, soil moisture is insufficient, which has a certain impact on crops and the ecological environment |
| $\leq -2.0$ | Extremely drought | 2.28 % | Soil moisture is seriously deficient for a long time, which has a serious impact on crops, ecological environment, industrial production as well as drinking water for people and animals |

ence scenario. Then, the probability distribution of change was calculated using CDFs.

## 3 Results

### 3.1 Spatial and temporal patterns of drought frequency and population

The frequency of mild, moderate, and extreme droughts, and their relationship with population were calculated for the reference period and the 1.5 °C global warming scenario to evaluate the spatial and temporal variation in frequency (Fig. S2) and population (Fig. S3). Generally, mild and moderate droughts will occur more frequently than extreme droughts. The frequency of mild and moderate droughts in most areas is in the range of 5 %–20 %, while the frequency of extreme droughts is less than 5 % in both the reference period and 1.5 °C global warming scenario (Fig. S2). As for the spatial pattern of frequency, areas with a high frequency of mild droughts are scattered, while moderate droughts are more spatially concentrated. In the reference period, moderate droughts are concentrated in southern China and the lower reaches of the Yellow River region, i.e., Beijing, Tianjin, Hebei, Henan, and Shandong province. In the 1.5 °C global warming scenario, the Shanxi-Shaanxi-Gansu-Ningxia region and Inner Mongolia-Xinjiang region also have more frequent moderate droughts. Extreme droughts occur primarily in inland areas. For example, the Qinghai-Tibet region has the highest frequency of extreme droughts in both scenarios. However, the spatial pattern of extreme droughts changed between the two scenarios. In the 1.5 °C global warming scenario, the frequencies in the northeast region, i.e., Heilongjiang, Jilin, and Liaoning provinces; the Shanxi-Shaanxi-Gansu-Ningxia region; and the middle and lower reaches of the Yangtze River region, i.e., Shanghai, Jiangsu, Anhui, Jiangxi Hunan, and Hubei province, decrease. In contrast, in the southwest region, i.e., Sichuan, Chongqing, Guizhou, and Yunnan provinces and the southeast coastal re-

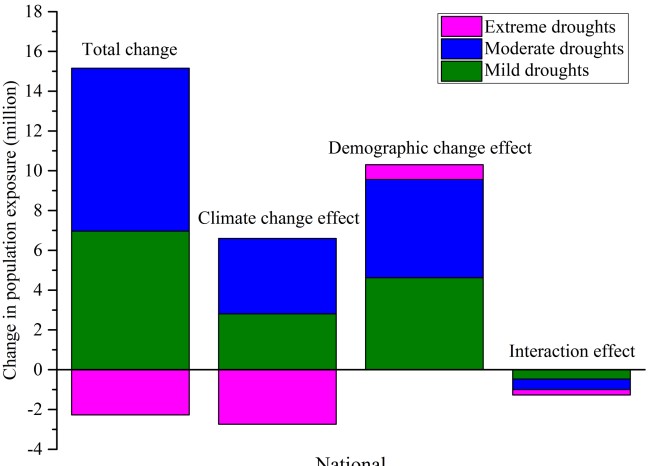

**Figure 4.** Decomposition of population exposure based on different effects – climate change, demographic change, and their interaction, as well as mild, moderate, and extreme droughts.

gion, i.e., Zhejiang, Fujian, Guangdong, Guangxi, Hainan, Hong Kong, Macao, and Taiwan, the frequency increases relative to the reference period.

The population of China increases by 32.56 million, from 1.26 billion in the reference period to 1.29 billion in the 1.5 °C global warming scenario. However, areas of increasing population do not expand in size, and most areas decrease. The variation in demographic change is clear when comparing the two sides of the Hu line (Hu, 1935), which is an imaginary line that diagonally divides the area of China into two parts, stretching from the city of Heihe in Heilongjiang Province to Tengchong in Yunnan Province. It is also called the "geo-demographic demarcation line"; the west of the line occupies 56.2 % of the area of China, but only 5.9 % of the population, while the east of the line occupies 43.8 % of the area, but 94.1 % of the population (Fig. S3). However, the spatial pattern of demographic change in number and percentage is different (Fig. S4). The number of peo-

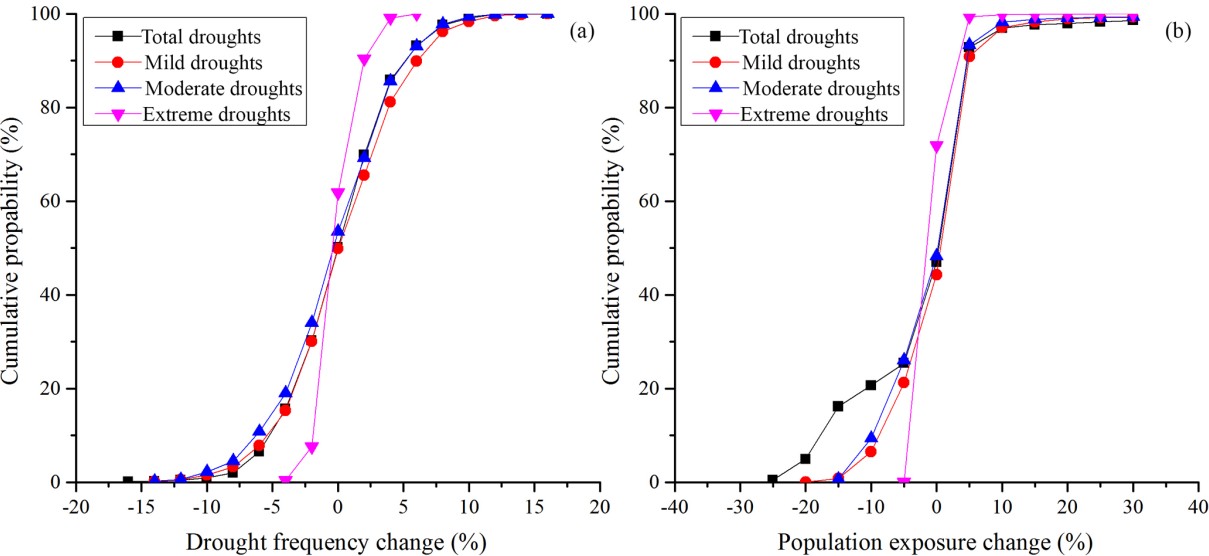

**Figure 5.** Cumulative probability projected change drought frequency **(a)** and population exposure **(b)** to mild, moderate, extreme, and total droughts.

ple decreases significantly on the east side of the Hu line, especially in the lower reaches of the Yellow River region and middle and lower reaches of the Yangtze River region, while the decrease in population by percentage is clear on the west side of the Hu line, such as the Inner Mongolia-Xinjiang region and east of the Qinghai-Tibet region. The reason for this dichotomy is the differing demographic distribution on both sides of the Hu line. The population density of the west side is so small that changes in percentage are clearer.

## 3.2   Spatial distribution and change in population exposure to droughts

The average annual aggregate population exposure in the reference period is 179.17 million and increases to 186.14 million in the 1.5 °C global warming scenario. Comparing the population exposure to different droughts in the reference period and the 1.5 °C global warming scenario (Fig. 1), the exposure to mild and moderate droughts increases while that to extreme droughts decreases. Moderate droughts account for 53.01 % of total exposure in the reference period and 53.34 % in the 1.5 °C global warming scenario, accounting for the most exposure. In comparison, mild droughts rank second and extreme droughts rank third accounting for 2.31 % and 1.69 % in the two scenarios. The spatial pattern of population exposure to droughts is similar to the population demographic distributions in China, i.e., divided by the Hu line. Exposure on the east side is much greater than on the west side (Fig. 2). Exposure in the Yellow River region and middle and lower reaches of the Yangtze River region is the highest, and it is lowest in the Inner Mongolia-Xinjiang region and the Qinghai-Tibet region.

Comparing the changes in exposure to mild (Fig. 3a), moderate (Fig. 3b), extreme (Fig. 3c), and total droughts (Fig. 3d), we found that, except for extreme droughts, they show similar spatial patterns. The exposure in southeast China increases, while that in the northwest part decreases. For mild droughts, exposure increases more clearly in the lower reaches of the Yellow River region, southeast region, and southeast coastal region. For moderate droughts, increases in the northeast region, Shanxi-Shaanxi-Gansu-Ningxia region, and southeast coastal region are apparent. In these regions, the combination of mild and moderate droughts dominates the overall pattern for total exposure. As for extreme droughts, the exposure for most of China decreases, except for the south of the southwest region and west of the southeast coastal region.

## 3.3   Relative importance analysis

The relative importance of different factors, i.e., climate change, demographic change, and interaction effects, as well as different droughts were analyzed (Fig. 4). For different factors, climate change and demographic change have positive impacts on the total exposure change (29.93 %, 79.95 %), while the interaction effect has a negative impact (−9.88 %). These results imply that the areas experiencing more droughts have decreasing populations in the 1.5 °C global warming scenario. For different droughts, the effect of mild and moderate droughts is positive and of similar magnitude, whereas extreme droughts have a lesser effect. Except for the constant climate scenario for analyzing the demographic change effect, the effect of extreme droughts is negative. In total change in exposure, the contributions from mild and moderate droughts are 54.03 % and 63.59 %, respec-

tively, leaving $-17.62\%$ for the effect of extreme droughts. In summary, the demographic change and moderate droughts are the dominant contributors to exposure change in the two scenarios.

## 3.4 Cumulative probability analysis

Figure 5 shows CDFs for drought frequency and population exposure for changes in the $1.5\,^\circ\mathrm{C}$ global warming scenario relative to the reference period. For the change in drought frequency (Fig. 5a), extreme droughts are in the minimum range, with changes of $-5\%$ to $5\%$, whereas total droughts are in the maximum range, $-18\%$ to $16\%$. The cumulative probabilities of an increase in drought frequency under the $1.5\,^\circ\mathrm{C}$ global warming scenario for mild, moderate, extreme, and total droughts are $50.14\%$, $46.48\%$, $38.23\%$, and $49.86\%$, respectively. Apart from extreme droughts, which show a clear downward trend, the probabilities of an increase or decrease in mild, moderate, and total droughts are roughly equal. In terms of change in population exposure (Fig. 5b), extreme droughts show a minimum, at $-5\%$ to $5\%$, and total droughts show a maximum, $-25\%$ to $25\%$, probability. Extreme droughts decrease, with a cumulative probability of $71.83\%$, while mild, moderate, and total droughts increase, with cumulative probabilities of $55.17\%$, $51.71\%$, and $53.01\%$, respectively. The probability of an increase in mild droughts is the highest, while the probability of an increase in extreme droughts is the lowest in both drought frequency and exposure under the $1.5\,^\circ\mathrm{C}$ global warming scenario.

## 4 Discussion

Extensive studies have focused on changes in extreme climate events under global climate change (Hirabayashi et al., 2013; Huang et al., 2017; Kharin et al., 2013). Currently, with the $1.5\,^\circ\mathrm{C}$ target, the socioeconomic impacts of $1.5\,^\circ\mathrm{C}$ global warming on factors, such as population exposed to disasters, need to be further studied. In this study, the population exposure to droughts was calculated for the $1.5\,^\circ\mathrm{C}$ global warming scenario and reference period by combining drought frequency and population simulations. The relative importance of different factors as well as evaluating the cumulative probability of exposure change were analyzed.

Our results indicate that average annual population exposure to droughts in the $1.5\,^\circ\mathrm{C}$ global warming scenario would increase by 6.97 million compared to the reference period, roughly $0.51\%$ of the projected Chinese population under the SSP1 scenario in 2030. The increase in exposure is rather unremarkable, suggesting that achieving the $1.5\,^\circ\mathrm{C}$ target may limit the potential damage incurred by climate change. Among the three different droughts, exposure to moderate droughts will be the largest because the areas with a high frequency of moderate droughts coincide with high population density. Drought frequency and population

are two important factors that contribute to exposure. To determine which one has a larger impact, the relative importance of these two factors, and the interaction effect, were analyzed. Results revealed that exposure change is mainly due to demographic change ($79.95\%$), and climate change is responsible for $29.93\%$ of the change; the interaction effects explain the remaining $-9.88\%$. These results are different from previous studies applying contribution analysis. Jones et al. (2015) calculated future population exposure to US heat extremes under the Special Report on Emission Scenarios (SRES) A2 scenario; they found that the growth in exposure is mainly due to climate change. Smirnov et al. (2016) analyzed the relative importance of climate change and demographic growth for exposure to future extreme droughts in RCP4.5 and RCP8.5, and their results indicated that climate change is more responsible for exposure change than demographic change in both scenarios. The contradiction may be due to the different scenarios used in the studies. SRES A2, RCP4.5, and RCP8.5 are scenarios with higher GHG emissions relative to RCP2.6, which corresponds to the $1.5\,^\circ\mathrm{C}$ target used in our study. This $1.5\,^\circ\mathrm{C}$ target is an important constraint because it is relevant to the requirements for large GHG emission reductions. The difference of GHG emissions also explains the cumulative probability analysis result for drought frequency. Drought frequency is not likely to increase in the $1.5\,^\circ\mathrm{C}$ global warming scenario compared to the reference period. Therefore, the effect of climate change to exposure is reduced compared to higher emission pathways, which results in demographic change acting as the primary contributor to exposure. In future studies, we would like to evaluate population exposure for high GHG emission pathways, i.e., RCP4.5/SSP2 and RCP8.5/SSP3, and compare with the results from RCP2.6/SSP1 to illustrate the impacts of achieving the $1.5\,^\circ\mathrm{C}$ target. Furthermore, studies accounting for more demographic characteristics in addition to growth, i.e., age, sex, education, and income would be carried out, which are likely to be stronger factors for demographic change in the $1.5\,^\circ\mathrm{C}$ target. However, we currently lack the required sophisticated data.

There are many kinds of droughts: meteorological, agricultural, hydrological, and socioeconomic. In this study, based on simulated climate data, we assessed population exposure to meteorological droughts under the $1.5\,^\circ\mathrm{C}$ global warming target using the SPEI; however, the results do not necessarily coincide with agricultural, hydrological, or socioeconomic droughts. Therefore, we would like to assess population exposure to different kinds of droughts to determine their impacts on populations. In addition, there are some uncertainties in estimating population exposure under climate change. The main sources include GHG emission scenarios (Maurer, 2007), GCMs (Kirono et al., 2011), calculating potential evapotranspiration, population prediction, and selection of the drought index (Burke and Brown, 2008). For instance, SPEI was chosen in this study because it combines the characteristics of SPI and PDSI; however, it is lim-

ited by providing a measure of dryness in a relative rather than absolute sense. Selecting different drought indexes may lead to differences in drought hazard and population exposure results. Therefore, future studies could evaluate different drought indexes based on more advanced and higher resolution GCMs and RCMs (regional climate models), determine importance of sources of uncertainty, and generate assessment results that are more accurate and reasonable.

## 5   Conclusions

This study leads to four key findings. First, population exposure to droughts on the east side of the Hu line is higher than that on the west side, which corresponds to general demographic distributions in China. Among the eight demographic regions, exposure in the middle and lower reaches of the Yangtze River region is the highest, and the lowest occurs in the Qinghai-Tibet region. Second, in the 1.5 °C global warming scenario, population exposure to droughts has a slight increase, 6.97 million more residents exposed, relative to the reference period. Third, variations in both population and climate are important factors in this change in exposure, but demographic change is the primary contributor (79.95 %) in the 1.5 °C global warming scenario. Moderate droughts contribute most among three droughts (63.59 %). Fourth, probabilities of increasing or decreasing total drought frequency are approximately equal (49.86 % and 49.66%, respectively), while the frequency of extreme drought is likely to decrease (71.83 % probability) in the 1.5 °C global warming scenario. To conclude, in the 1.5 °C global warming scenario, the contribution of climate change is significantly less than demographic change and drought frequency will not increase distinctly compared to the reference period, which indicates that reaching the 1.5 °C target is a potential mechanism for mitigating the impact of climate change on both droughts and population exposure. In addition, demographic change should be regarded as a significant component to control the growth in exposure to droughts.

**Data availability.**  TS2 Data supporting this study are available in public repositories. The meteorological data are available from https://www.isimip.org/ (last access: 20 August 2018). The population data are available from http://www.cger.nies.go.jp/gcp/population-and-gdp.html (last access: 20 August 2018).

**The Supplement related to this article is available online at https://doi.org/10.5194/esd-9-1-2018-supplement.**

**Author contributions.**  JC and YL conceived the study. JC performed all analysis and wrote the initial manuscript with support from YL and TP. All authors contributed to interpreting results, discussion, and improvement of the manuscript.

**Competing interests.**  The authors declare that they have no conflict of interest. TS3

**Special issue statement.**  This article is part of the special issue "The Earth system at a global warming of 1.5 and 2.0 °C". It is not associated with a conference.

**Acknowledgements.**  This study was supported by the National Key Research and Development Program of China (grant no. 2016YFA0602402); the National Natural Science Foundation of China (grant nos. 41671037, 41301091); the Key Research Program of Frontier Sciences, CAS (QYZDB-SSW-DQC005); and the Youth Innovation Promotion Association, CAS (grant no. 2016049). We also thank ISI-MIP and NIES for data support.

Edited by: Axel Kleidon
Reviewed by: four anonymous referees

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

## Remarks from the typesetter