# Peer review of "Population exposure to droughts in China under 1.5 °C global warming target"

_Earth System Dynamics, 2017_

## Referee Comment (RC1) · Anonymous Referee #1 · 30 Nov 2017

The "Population exposure to droughts in China under 1.5 °C global warming target" has important implications for the adaption and coping with climate change. Overall, this study is well motivated, and is generally well-written. I have only few relatively minor comments, and therefore I recommend to return the manuscript to the authors for minor revisions. My main comments and suggestions are listed below.

Please define "risk" and "exposure" in introduction.

Rephrase P3 Line 12 to 13.

Please give more details on SPEI calculation.

Please define the "Hu line" and provide a brief introduction.

[Figure]

P5 Line 5, two scenarios?

Figure 5, figure (a) and figure (b) almost the same, so I suggest to add a total number of population.

P7 "Results suggest that reaching the 1.5 °C target is a potential mechanism for mitigating the impact of climate change on droughts." It is not very clear.

---

## Author Comment (AC1) · 15 Dec 2017

Dear Editors and Reviewers:

Thank you for your letter and for the reviewer's comments concerning our manuscript entitled "Population exposure to droughts in China under 1.5°C global warming target" (ID: esd-2017-100). Those comments are all valuable and very helpful for revising and improving our manuscript. We studied comments carefully and made corrections in the manuscript. The response to the reviewer's comments are as follow:

1. Please define "risk" and "exposure" in introduction.

Authors' response: Thanks for your suggestions. We have supplemented the definition of "risk" and "exposure" in introduction in P1 Line 30. The statement is: "Risk is often

represented as probability of occurrence of hazardous events or trends multiplied by the impacts if these events or trends occur, it results from the interaction of hazard, exposure, and vulnerability (Field et al., 2014). Therefore, Exposure assessment is one of the most important aspects of disaster risk assessment. Exposure usually refers to the presence of people, livelihoods, species or ecosystems, environmental functions, services, and resources, infrastructure, or economic, social, or cultural assets in places and settings that could be adversely affected (Field et al., 2014)."

2. Rephrase P3 Line 12 to 13.

Authors' response: Thanks for your advice. The statement was rephrased to "Combined the characteristics of the Standardized Precipitation Index (McKee et al., 1993) with multi-scale and Palmer Drought Severity Index (Palmer, 1965) which is sensitive to warming, SPEI was proposed by Vicente-Serrano et al. (2010)."

3. Please give more details on SPEI calculation.

Authors' response: Thank you. We have added the statement "SPEI reflects the change of water defect by using the Log-logistic probability distribution function, and obtains the drought index value by normalized normalization. In this study, SPEI is calculated by R-SPEI-package. The radiation coefficient used is based on the radiation calibration results in China by Yin et al. (2008)." in Part 2.2 Calculation of SPEI in P3.

4. Please define the "Hu line" and provide a brief introduction

Authors' response: Thanks for your suggestions. We have supplemented the definition and a brief introduction of "Hu line". The statement is "Hu line is an imaginary line that divides the area of China into two parts, stretching from the city of Heihe in Heilongjiang province to Tengchong in Yunnan province, diagonally across China. It is also called "geo-demographic demarcation line", the west of the line occupies 64% of the area of China, but only 4% of the population while the east of the line occupies 36% of the area, but 96% of the population (Hu 1935)." In addition, we have added the Hu line in

Fig 2, Fig. S3 and S4 so that the statement and figures would be easily understood.

5. P5 Line 5, two scenarios?

Authors' response: Sorry for our incorrect writing of "two scenarios" in P5 Line 5. We have corrected the statement to "reference period and 1.5°C global warming scenario".

6. Figure 5, figure (a) and figure (b) almost the same, so I suggest to add a total number of population.

Authors' response: Thanks for your suggestions. Figure 5 shows the cumulative probability projected change drought frequency and population exposure in order to reflect the change of frequency and exposure of the three grades of droughts. Therefore, the change of population is not included. Figure (a) and figure (b) are similar because most of the probabilities of increase in frequency (a) and exposure (b) are near 50%, but there are some differences between the two figures. For example, the probability of decrease of extreme droughts in frequency and exposure is 61.77% and 71.83 % respectively. Besides, we have shown the change of population both in number and percentage in Fig. S4. Of course we also think the number of population is important, the suggestion is valuable, so we added the spatial distribution of population of China in reference period in Fig. S3.

7. P7 "Results suggest that reaching the 1.5°C target is a potential mechanism for mitigating the impact of climate change on droughts." It is not very clear.

Authors' response: Thanks for your comments. The statement is based on the results from our study. To make it more clear, we have rewritten this statement to "Fourthly, probabilities of increase or decrease in drought frequency are roughly equal (49.86 % and 49.66% respectively) while the frequency of extreme drought is probably to decrease (71.83 % probability) in 1.5°C global warming scenario. Results suggest that in 1.5°C global warming scenario, the contribution of climate change is significantly less than demographic change and drought frequency has not increased obviously com-

pared to reference period. Therefore, reaching the 1.5°C target is a potential mechanism for mitigating the impact of climate change on both droughts and population exposure."
* * *
[Figure]

**Fig. 1.** Figure 2. Spatial distribution of population exposure to droughts in (a) the reference period (1986-2005) and (b) 1.5 °C global warming scenario (2020-2039 in RCP2.6).

[Figure]

**Number of Population**

| | |
|---|---|
| $<1$ | $10^4 - 10^5$ |
| $1 - 10^3$ | $10^5 - 10^6$ |
| $10^3 - 10^4$ | $>10^6$ |

**Fig. 2.** Fig. S3 Spatial distribution of population and geo-demographic demarcation line (Hu line) of China in reference period.

[Figure]

**Fig. 3.** Fig. S4 Change in population, in number (a) and percentage (b), between the reference period and 1.5 °C global warming scenario.

---

## Referee Comment (RC2) · Anonymous Referee #2 · 24 Jan 2018

First, thank-you for the opportunity to reveiw this paper. I found it interesting and thought-provoking. Second, my apologies for the slow review on my part.

In general I think this is a timely, well organized, and well-written study that addresses a significant consequence of climate change in human terms, but within the bounds of the stated goal of the Paris Accord, limiting global temperature rise to 1.5C above per-industrial. The piece adds to the small but rapidly growing literature that considers demographic and socioeconomic change alongside the physical consequences of climate change. Of particular interest is the disaggregation of the total increase in exposure into the climate, population, and interaction components.

While I believe the study is very well conceived and the paper is very well written, I have to object to one of the authors primary conclusions. I do not believe an increase

in exposure of 6.97 million persons constitutes a "substantial" increase. If anything I would argue that it is quite the opposite. In 2030 6.97 million persons represent roughly 0.5% of the projected Chinese population under SSP1 (1.359 billion). In short, as currently contextualized, the results/projections are a bit misleading as the increase in exposure is rather unremarkable.

I would suggest two possible pathways to remedy this issue. First, the authors might reframe this result to highlight the importance of achieving the goals of the Paris Accord within the context of Chinese droughts. This study finds that doing so will limit the potential damage incurred by climate change. Second, this finding might be supported by adding an additional scenario, such as an SSP2/RCP4.5, SSP3/RCP4.5, or SSP5/RCP8.5 combination to illustrate the avoided impacts of achieving Paris. The second suggestion entails significantly more work, and may be better thought of as future work, but at the very least I would like to see the paper reframed to better fit with the results.

I have only two other minor comments:

Two Minor points:

Page 3 lines 27-29: I would suggest rewriting as "The impact of population was calculated by holding climate constant, that is, the frequency of mild, moderate, and extreme droughts in the reference period multiplied by the population in the SSP1 scenario" (as opposed to ..."the population in the 1.5C global warming scenario). You want to convey to the reader that you are holding climate constant and allowing population to vary, so use the SSP as opposed to the temperature target.

Page 4 line 4: I am assuming exposure is expressed in "average annual" population counts. I would suggest adding this terminology up front in Section 3.2 (e.g., "The average annual aggregate exposure.....)

---

## Referee Comment (RC3) · Anonymous Referee #3 · 1 Feb 2018

Unfortunately, I do not share the positive view of the other two reviewers about this manuscript. I find the idea of the study interesting and worth exploring. But the analysis suffers from a number of severe flaws and therefore fails to provide meaningful quantifications and insights. Both the analysis and the text would require major and fundamental revisions to come to a publishable manuscript. My points of concern are as follows:

1. The standardized precipitation evapotranspiration index (SPEI) used in this study is an index of meteorological drought. Meteorological droughts do not necessarily coincide with agricultural, hydrological, or even socio-economic drought (see Wilhite, D. A. and Glantz, M. H. (1985) 'Understanding the Drought Phenomenon: The Role of Definitions', Water International, 10(3), pp. 111-120. doi: 10.1080/02508068508686328).

Thus, meteorological droughts have only limited direct relevance to people. In addition, the SPEI defines meteorological drought as departure from the mean climatic water balance (precipitation minus potential evapotranspiration) in multiples of standard deviations. For example, a value of -1 marks an event that deviates by one standard deviation from mean conditions. By definition, 15.9% of all time steps will be classified as -1 or less. It is obvious that such an indicator does not provide a measure of dryness in an absolute sense. Under wet conditions with low temporal variability, most SPEI droughts are still wet in an absolute sense; under dry conditions, many very dry events may not be classified as drought by the SPEI. Despite these shortcomings, I do believe that assessing population exposure to changes in meteorological droughts under climate change is a valid research question. But the limitations of the employed indicator (and drought type) must be highlighted and discussed to avoid misinterpretation of the results. This is clearly lacking in the paper, which instead tends to overstate the meaning of population exposure to meteorological droughts (e.g., page 2, lines 8-11).

2. The basic concept of the SPEI is to transform a time series of the climatic water balance into a time series of normally distributed index values with a mean of 0 and a standard deviation of 1. For this transformation, a probability distribution function is fitted to the empirical distribution of climatic water balance values. The fitted distribution function is then used to map the climatic water balance values to SPEI values corresponding to the same quantile. Performing the transformation for present day and future time periods with independently fitted distribution functions, will yield two SPEI time series with the same statistical properties. Any attempt to identify a climate change signal will fail with this approach as the signal is lost in the transformation. Therefore, a single distribution function (preferably estimated from the reference period) must be used for the transformation of both the reference and future time series to be able to detect changes in the frequency of drought events. It is not clear whether this has been done correctly in this analysis as the method sections only provides a very vague description of the SPEI calculation. However, the results and how they are

presented indicate that separate distribution functions have been fitted to the reference and the future time period.

3. On page 2 line 32 the authors explain that the climate data from the five available GCMs had been averaged prior to the analysis. Averaging time series is never a good idea. But in the case of GCM time series and with the aim to calculate SPEI it is simply wrong. The argument that "combining multiple models has been to shown superior to a single model" only holds true for long term averages and only for the comparison to observations. The SPEI analysis must be performed for each GCM individually. The results can then be averaged while properly accounting for GCM uncertainty.

4. The paper defines population exposure to drought as "the frequency of mild, moderate, and extreme droughts multiplied by the number of people exposed to them" and reports it as number of people. I don't think this is appropriate. Let's assume a moderate drought is found to occur over 10 % of the time in a given grid cell. Then, according to the above definition, 10 % of the total population in that grid cell would be counted as exposed to moderate drought. This is strange because intuitively one would expect that all people in that cell will experience moderate drought conditions over 10 % of the time. It is possible that it is only the unit (population numbers) that is puzzling here and that it could be fixed by including the temporal dimension. However, under no circumstance should the population exposure obtained for different drought severity classes be added (as done on multiple occasions in the paper).

5. The methods description is very short and lacks explanation of important aspects, which are crucial for the understanding of the analysis. It is by no means clear how ETo was calculated (e.g., climate variables used, temporal resolution) and which procedure was used to derive the SPEI (e.g., temporal resolution or number of time steps of SPEI, probability distribution type assumed for climatic water balance, fitting methods for estimating parameters probability distribution function, same or different parameters for reference period and scenario). In order to assure transparency and reproducibility of the analysis this information must be provided.

6. It is not clear to me how the section 3.4 can contribute to a quantification of uncertainties.

---

## Author Comment (AC2) · 28 Feb 2018

Dear Editors and Reviewers: Thank you for your letter and for the reviewer's comments concerning our manuscript entitled "Population exposure to droughts in China under 1.5°C global warming target" (ID: esd-2017-100). Those comments are all valuable and very helpful for revising and improving our manuscript. We studied comments carefully and made corrections in the manuscript. The response to the reviewer's comments are as follow:

1. While I believe the study is very well conceived and the paper is very well written, I have to object to one of the author's primary conclusions. I do not believe an increase in exposure of 6.97 million persons constitutes a "substantial" increase. If anything

[Figure]

I would argue that it is quite the opposite. In 2030 6.97 million persons represent roughly 0.5% of the projected Chinese population under SSP1 (1.359 billion). In short, as currently contextualized, the results/projections are a bit misleading as the increase in exposure is rather unremarkable. I would suggest two possible pathways to remedy this issue. First, the authors might reframe this result to highlight the importance of achieving the goals of the Paris Accord within the context of Chinese droughts. This study finds that doing so will limit the potential damage incurred by climate change. Second, this finding might be supported by adding an additional scenario, such as an SSP2/RCP4.5, SSP3/RCP4.5, or SSP5/RCP8.5 combination to illustrate the avoided impacts of achieving Paris. The second suggestion entails significantly more work, and may be better thought of as future work, but at the very least I would like to see the paper reframed to better fit with the results.

Authors' response: Thanks for your suggestions. The statement was rephrased in Section 4 and 5. The modifications are as follow:

P7 Line 16-18, We have added the statement "(The results indicated that average annual population exposure to droughts in the 1.5 °C global warming scenario would increase by 6.97 million compared to the reference period,) roughly 0.51 % of the projected Chinese population under the SSP1 scenario in 2030. The increase in exposure is rather unremarkable, suggesting that achieving the 1.5 °C target may limit the potential damage incurred by climate change."

P8 Line 3-5, We have added the statement "In future studies, we would like to evaluate population exposure for high GHG emission pathways, i.e., RCP4.5/SSP2 and RCP8.5/SSP3, and compare with the results from RCP2.6/SSP1 to illustrate the impacts of achieving the 1.5 °C target."

P8 Line 24 We have revised the statement "a substantial increase" to "a slight increase".

2. Page 3 lines 27-29: I would suggest rewriting as "The impact of population was

calculated by holding climate constant, that is, the frequency of mild, moderate, and extreme droughts in the reference period multiplied by the population in the SSP1 scenario" (as opposed to ..."the population in the 1.5C global warming scenario). You want to convey to the reader that you are holding climate constant and allowing population to vary, so use the SSP as opposed to the temperature target.

Authors' response: Thanks for your advice. The statement was rewritten to "The impact of population was calculated by holding climate constant, that is, the frequency of mild, moderate, and extreme droughts in the reference period multiplied by the population in the SSP1 scenario. Similarly, when calculating impact of climate, the population was held constant, that is, the frequency of mild, moderate, and extreme droughts in the RCP2.6 scenario was multiplied by the population in the reference period."

3. Page 4 line 4: I am assuming exposure is expressed in "average annual" population counts. I would suggest adding this terminology up front in Section 3.2 (e.g., "The average annual aggregate exposure.....)

Authors' response: Thanks for your suggestions. We have added the express "average annual (aggregate exposure) " in Page 6 line 2(Section 3.2 ) as well as Page 7 line 15(Section 4 ).

---

## Author Comment (AC3) · 28 Feb 2018

Dear Editors and Reviewers: Thank you for your letter and for the reviewer's comments concerning our manuscript entitled "Population exposure to droughts in China under 1.5°C global warming target" (ID: esd-2017-100). Those comments are all valuable and very helpful for revising and improving our manuscript. We studied comments carefully and made corrections in the manuscript. The response to the reviewer's comments are as follow:

1. The standardized precipitation evapotranspiration index (SPEI) used in this study is an index of meteorological drought. Meteorological droughts do not necessarily coincide with agricultural, hydrological, or even socio-economic drought Thus, meteo-

rological droughts have only limited direct relevance to people. In addition, the SPEI defines meteorological drought as departure from the mean climatic water balance (precipitation minus potential evapotranspiration) in multiples of standard deviations. For example, a value of -1 marks an event that deviates by one standard deviation from mean conditions. By definition, 15.9% of all time steps will be classified as -1 or less. It is obvious that such an indicator does not provide a measure of dryness in an absolute sense. Under wet conditions with low temporal variability, most SPEI droughts are still wet in an absolute sense; under dry conditions, many very dry events may not be classified as drought by the SPEI. Despite these shortcomings, I do believe that assessing population exposure to changes in meteorological droughts under climate change is a valid research question. But the limitations of the employed indicator (and drought type) must be highlighted and discussed to avoid misinterpretation of the results. This is clearly lacking in the paper, which instead tends to overstate the meaning of population exposure to meteorological droughts (e.g., page 2, lines 8-11).

Authors' response: Thanks for your suggestions. We have added the express about limitations of the employed indicator (and drought type) in Section 4. The modifications are as follow:

P8 Line 8-11, we have added the statement "There are many kinds of droughts: meteorological, agricultural, hydrological, and socioeconomic. In this study, based on simulated climate data, we assessed population exposure to meteorological droughts under the 1.5 °C global warming target using the SPEI; however, the results do not necessarily coincide with agricultural, hydrological, or socioeconomic droughts. Therefore, we would like to assess population exposure to different kinds of droughts to determine their impacts on populations."

P8 Line 14-19, we have added the statement "For instance, SPEI was chosen in this study because it combines the characteristics of SPI and PDSI; however, it is limited by providing a measure of dryness in a relative rather than absolute sense. Selecting different drought indexes may lead to differences in drought hazard and population

exposure results. Therefore, future studies could evaluate different drought indexes based on more advanced and higher resolution GCMs and RCMs (regional climate models), determine importance of sources of uncertainty, and generate assessment results that are more accurate and reasonable."

2. The basic concept of the SPEI is to transform a time series of the climatic water balance into a time series of normally distributed index values with a mean of 0 and a standard deviation of 1. For this transformation, a probability distribution function is fitted to the empirical distribution of climatic water balance values. The fitted distribution function is then used to map the climatic water balance values to SPEI values corresponding to the same quantile. Performing the transformation for present day and future time periods with independently fitted distribution functions, will yield two SPEI time series with the same statistical properties. Any attempt to identify a climate change signal will fail with this approach as the signal is lost in the transformation. Therefore, a single distribution function (preferably estimated from the reference period) must be used for the transformation of both the reference and future time series to be able to detect changes in the frequency of drought events. It is not clear whether this has been done correctly in this analysis as the method sections only provides a very vague description of the SPEI calculation. However, the results and how they are presented indicate that separate distribution functions have been fitted to the reference and the future time period.

Authors' response: Thanks for your advice. We have added detailed statements of SPEI calculation in Section 2.2 and supplemented probability distribution of SPEI for different drought grades in Table 1.

3. On page 2 line 32 the authors explain that the climate data from the five available GCMs had been averaged prior to the analysis. Averaging time series is never a good idea. But in the case of GCM time series and with the aim to calculate SPEI it is simply wrong. The argument that "combining multiple models has been to shown superior to a single model" only holds true for long term averages and only for the comparison to

observations. The SPEI analysis must be performed for each GCM individually. The results can then be averaged while properly accounting for GCM uncertainty.

Authors' response: Thanks for your comments. Our inappropriate description led to a misunderstanding of the analysis. We have calculated SPEI for each GCM initially and averaged the results for drought frequency and population exposure analysis. So we have replaced the description to "In this study, we synthesized the results of the five GCMs based on the separately calculated SPEI for each GCM, as combining results of multiple models has been shown to be superior to a single model (Zhou and Yu, 2006)." in Page 3 line 6-8 (Section 2.1). Besides, we have added uncertainty discussion including GCM uncertainty in Page 8 line 12-19 (Section 4). The statement was rephrased to"In addition, there are some uncertainties in estimating population exposure under climate change. The main sources include GHG emission scenarios (Maurer, 2007), GCMs (Kirono et al., 2011), calculating potential evapotranspiration, population prediction, and selection of the drought index (Burke and Brown, 2008). For instance, SPEI was chosen in this study because it combines the characteristics of SPI and PDSI; however, it is limited by providing a measure of dryness in a relative rather than absolute sense. Selecting different drought indexes may lead to differences in drought hazard and population exposure results. Therefore, future studies could evaluate different drought indexes based on more advanced and higher resolution GCMs and RCMs (regional climate models), determine importance of sources of uncertainty, and generate assessment results that are more accurate and reasonable."

4. The paper defines population exposure to drought as "the frequency of mild, moderate, and extreme droughts multiplied by the number of people exposed to them" and reports it as number of people. I don't think this is appropriate. Let's assume a moderate drought is found to occur over 10 % of the time in a given grid cell. Then, according to the above definition, 10 % of the total population in that grid cell would be counted as exposed to moderate drought. This is strange because intuitively one would expect that all people in that cell will experience moderate drought conditions over 10 % of the

time. It is possible that it is only the unit (population numbers) that is puzzling here and that it could be fixed by including the temporal dimension. However, under no circumstance should the population exposure obtained for different drought severity classes be added (as done on multiple occasions in the paper).

Authors' response: Thanks for your comments. There are different kinds of definition of population exposure to extreme climate events and disasters. For example, Smirnov et.al (2016) defined "populations' exposure to extreme drought as the total number of people, in the world or in a country, living in grid cells where SPEI $< -2$." While the definition of exposure we used is referred to Jones et al (2015), which defined population exposure to heat extremes as "the annual average number of days with a maximum temperature above 35 °C multiplied by the number of people exposed to that outcome." To state more clearly, we have change the description to "Our measure of population exposure is the number of people exposed to mild, moderate, and extreme droughts. That is, the annual average percentage of mild, moderate, and extreme droughts multiplied by the number of people exposed to that outcome, which is referred to Jones et al. (2015)" in Page 4 line 9-11 (Section 2.3).

As for calculation of population exposure of different severity classes, it is referred to studies of Smirnov et al. (2016) and Sun et al. (2017) as is mentioned in Section 1, which is also widely used in relevant studies. Smirnov et al. (2016) assessed population exposure to extreme droughts while the study did not account for mild and moderate droughts. Sun et al. (2017) analyzed population exposure to moderate, severe and extreme droughts under 1.5 °C and 2.0 °C global warming scenarios, while the study ignored the impact of demographic growth on population exposure change. In this study, calculation of population exposure of different severity classes make the results more accurate, and is useful for relative importance analysis. In addition, it is also important for vulnerability and risk assessment in further studies.

References:

Jones, B., O'Neill, B. C., Mcdaniel, L., Mcginnis, S., Mearns, L. O., and Tebaldi, C.: Future population exposure to US heat extremes, Nat. Clim. Change, 5, 592–597, 2015.

Smirnov, O., Zhang, M., Xiao, T., Orbell, J., Lobben, A., and Gordon, J.: The relative importance of climate change and population growth for exposure to future extreme droughts, Climatic Change, 138, 1–13, 2016.

Sun, H., Wang, Y., Chen, J., Zhai, J., Jing, C., Zeng, X., Ju, H., Zhao, N., Zhan, M., and Luo, L.: Exposure of population to droughts in the Haihe River Basin under global warming of 1.5 and 2.0°C scenarios, Quatern. Int., 2017.

5. The methods description is very short and lacks explanation of important aspects, which are crucial for the understanding of the analysis. It is by no means clear how ETo was calculated (e.g., climate variables used, temporal resolution) and which procedure was used to derive the SPEI (e.g., temporal resolution or number of time steps of SPEI, probability distribution type assumed for climatic water balance, fitting methods for estimating parameters probability distribution function, same or different parameters for reference period and scenario). In order to assure transparency and reproducibility of the analysis this information must be provided.

Authors' response: Thanks for your suggestions. We have supplemented detailed calculation process of ET0 in Section 2.2 including climate variables used and temporal resolution. Also, we added the procedure used to derive the SPEI and the set parameters. The statement is:

Page 3 line 23-24, "Differences between precipitation (P) and potential evapotranspiration (ET0), which reflect the water surplus or deficit in a region, were calculated to deduce the SPEI using:"

Page 3 line 27-30, "Therefore, the Penman–Monteith equation (FAO, 1998) was replaced to calculate ET0 in this study. The Penman–Monteith equation comprehensively

considers the impact of both thermal and dynamic factors on ET0, i.e., temperature, wind speed, relative humidity, and solar radiation. Therefore, results are more consistent with true reference crop evapotranspiration."

Page 4 line 2-7, "Here, ET0 is the potential evapotranspiration; Rn is the net radiation; G is the soil heat flux density; T is the surface mean daily air temperature; u2 is the wind speed at 2 m height above the ground; es is the saturation vapor pressure; and ea is the actual vapor pressure. The SPEI was calculated using the R-SPEI-package (https://CRAN.R-project.org/package=SPEI). The input data are monthly time series of D (differences between precipitation and potential evapotranspiration), where the set parameters are scale=12, kernel = 'rectangular', distribution = 'log-Logistic', and fit = 'ub-pwm'. The categorization of drought grade by SPEI and its probability are shown in Table 1 (Liu and Jiang, 2015)."

6. It is not clear to me how the section 3.4 can contribute to a quantification of uncertainties.

Authors' response: Thanks for your comments. In this study, cumulative distribution functions (CDFs) were used to quantified drought frequency and population exposure change in 1.5 °C global warming scenario relative to reference period to evaluate the possible impact of climate change. Uncertainty analysis in Section 2.4 refers to uncertainty analysis of drought frequency and population exposure change, the statement may lead to misunderstanding. Thus, we have replaced the title of Section 2.4 to "Relative importance and cumulative probability analysis" and the definition of CDFs was added in Page 4 line 23-24 (Section 2.4), the statement is "The CDF of a random variable X is the function representing the probability that the random variable X takes on a value less than or equal to x".

---

## Author Comment (AC4) · 6 Mar 2018

Response to reviewer #1

Dear Editors and Reviewers:

Thank you for your letter and for the reviewer's comments concerning our manuscript entitled "Population exposure to droughts in China under 1.5°C global warming target" (ID: esd-2017-100). Those comments are all valuable and very helpful for revising and improving our manuscript. We studied comments carefully and made corrections in the manuscript. The response to the reviewer's comments are as follow:

1. Please define "risk" and "exposure" in introduction.

Authors' response: Thanks for your suggestions. We have supplemented the definition of "risk" and "exposure" in P1 Line 30 (Section 1 Introduction). The statement is: "Risk is often represented as the probability of occurrence of hazardous events or trends multiplied by the impacts if these events or trends occur, it results from the interaction of hazard, exposure, and vulnerability (Field et al., 2014). Therefore, exposure assessment is one of the most important aspect of disaster risk assessment. Exposure usually refers to the presence of people, livelihoods, species or ecosystems, environmental functions, services, and resources, infrastructure, or economic, social, or cultural assets in places and settings that could be adversely affected (Field et al., 2014)."

2. Rephrase P3 Line 12 to 13.

Authors' response: Thanks for your advice. The statement was rephrased to "Combined the characteristics of the Standardized Precipitation Index (SPI) (McKee et al., 1993) at multiple scales and Palmer Drought Severity Index (PDSI) (Palmer, 1965) which is sensitive to warming, SPEI was proposed by Vicente-Serrano et al. (2010)."

3. Please give more details on SPEI calculation.

Authors' response: Thank you. We have added the statement "The SPEI reflects the change in water deficit using the Log-logistic probability distribution function, and obtains the drought index value by normalized normalization." and "The radiation coefficient used is based on the radiation calibration results in China provided by Yin et al. (2008)". In Addition, detailed calculation process of potential evapotranspiration as well as procedure used to derive the SPEI and the set parameters were also supplemented in Section 2.2.

4. Please define the "Hu line" and provide a brief introduction

Authors' response: Thanks for your suggestions. We have supplemented the definition and a brief introduction of "Hu line" in Section 3.1. The statement is "(The variation in

demographic change is clear when comparing the two sides of the Hu line), which is an imaginary line that diagonally divides the area of China into two parts, stretching from the city of Heihe in Heilongjiang Province to Tengchong in Yunnan Province. It is also called the "geo-demographic demarcation line"; the west of the line occupies 56.2 % of the area of China, but only 5.9 % of the population, while the east of the line occupies 43.8 % of the area, but 94.1 % of the population (Fig. S3)." In addition, we have added the Hu line in Fig 2, Fig S3 and S4 so that the statement and figures would be easily understood.

5. P5 Line 5, two scenarios?

Authors' response: Sorry for our incorrect writing of "two scenarios". We have corrected the statement to "the reference period and the 1.5°C global warming scenario".

6. Figure 5, figure (a) and figure (b) almost the same, so I suggest to add a total number of population.

Authors' response: Thanks for your suggestions. Figure 5 shows the cumulative probability projected change drought frequency and population exposure in order to reflect the change of frequency and exposure of the three grades of droughts. Therefore, the change of population is not included. Figure (a) and figure (b) are similar because most of the probabilities of increase in frequency (a) and exposure (b) are near 50%, but there are some differences between the two figures. For example, the probability of decrease of extreme droughts in frequency and exposure is 61.77% and 71.83 % respectively. Besides, we have shown the change of population both in number and percentage in Fig S4. Of course we also think the number of population is important, the suggestion is valuable, so we added the spatial distribution of population of China in reference period in Fig S3.

7. P7 "Results suggest that reaching the 1.5°C target is a potential mechanism for mitigating the impact of climate change on droughts." It is not very clear.

Authors' response: Thanks for your comments. The statement is based on the results from our study. To make it more clear, we have rewritten this statement to "Fourth, probabilities of increasing or decreasing total drought frequency are approximately equal (49.86 % and 49.66% respectively), while the frequency of extreme drought is likely to decrease (71.83 % probability) in 1.5°C global warming scenario. Results suggest that in the 1.5°C global warming scenario, the contribution of climate change is significantly less than demographic change and drought frequency will not increase distinctly compared to reference period, which indicates that reaching the 1.5°C target is a potential mechanism for mitigating the impact of climate change on both droughts and population exposure." in Section 5.

[Figure]

[Figure]

**Fig. 1.** Figure 2. Spatial distribution of population exposure to droughts in (a) the reference period (1986-2005) and (b) 1.5 °C global warming scenario (2020-2039 in RCP2.6).

**Number of Population**

| | |
|---|---|
| <1 | $10^4$-$10^5$ |
| 1-$10^3$ | $10^5$-$10^6$ |
| $10^3$-$10^4$ | >$10^6$ |

the South
China Sea
Islands

Heihe

Tengchong

**Fig. 2.** Fig. S3 Spatial distribution of population and geo-demographic demarcation line (Hu line) of China in reference period.

[Figure]

[Figure]

**Fig. 3.** Fig. S4 Change in population, in number (a) and percentage (b), between the reference period and 1.5 °C global warming scenario.

---

## Author Comment (AC5) · 6 Mar 2018

Attached the marked up version manuscript.

Relative to the old manuscript, substantial revisions were made in this version based on the comments/suggestions from the editor and three reviewers. The texts mark in red, blue and purple are modifications according to reviewer #1, 2 and 3 respectively. We thank the editor and anonymous reviewers for their helpful comments.

Please also note the supplement to this comment:
https://www.earth-syst-dynam-discuss.net/esd-2017-100/esd-2017-100-AC5-supplement.pdf

[Figure]

**Supplement:**

[revised manuscript text omitted]

---

## Author Response (AR2)

Response to reviewer #4

Dear Editors and Reviewers:

Thank you for your letter and for the reviewer's comments concerning our manuscript entitled "Population exposure to droughts in China under 1.5°C global warming target" (ID: esd-2017-100). Those comments are all valuable and very helpful for revising and improving our manuscript. We studied comments carefully and made corrections in the manuscript. The response to the reviewer's comments are as follow:

1. Remarkably, it is not clear what is the period that the authors actually study. The authors refer to the period as 1.5 C global warming scenario period but never define it explicitly. Is it 2020-2040, 2040-2060, 2050-2100, 2080-2100, etc.?? This is especially importance since the authors compare their results with other studies, which may cover different periods. In one place, the authors mention population projection for 2030 -- perhaps this is the period that they study. In that case, the authors should strongly emphasize that their results are for 2030 and not for other periods. For example, the population growth for China is generally expected to be non-linear; i.e., it is likely to decrease by the end of the century.

**Authors' response:** *Thanks for your suggestions. The period of 1.5 C global warming scenario was determined in Page 3 line 10-11(Section 2.1). The statement was "According to previous research (Schleussner et al., 2016; Sun et al., 2017), a stable increase of 1.5°C GMT above preindustrial level for 20 years will be in 2020-2039 under RCP2.6". Of course we also think the study period should be emphasized to avoid misunderstanding, the suggestion is valuable, so we added the time period of both reference period (1986-2005) and 1.5 C global warming scenario (2020-2039 in RCP2.6) in Page 1 line 14 (Abstract) 、Page 4 line 15-16 (Section 2.3)、Page 12 line 6(caption of Figure 1)、Page 13 line 5-6(caption of Figure 3)and Supplement(captions of Fig. S2、Fig. S3 and Fig. S4).*

2. The results are entirely based on the SSP data. While I do not suggest replicating the drought exposure results for other population growth projections, the authors should at least compare SSP projections with other popular projections such as the United Nations scenarios.

**Authors' response:** *Thanks for your suggestions. The reason for using SSP data rather than other popular population projections in this study is the correspondence between RCP and SSP scenarios. SSP1 we chose is a sustainable development scenario facing low mitigation and adaption challenges which may be closer to 1.5°C global warming target. While other population projections seems not consider the specific carbon emission concentration or target of increase in global mean temperature in the future. Of course we also think comparing SSP projections with other popular projections is important for make our choice more reasonable, so a brief comparing was added in Page 3 line 11-15(Section 2.1). The statement was "As for population data, the United*

*Nations (http://www.un.org), world bank (https://data.worldbank.org.cn) and other organizations have proposed projection of population in the future. Considering the RCP scenario we chose in this study, projections of population data in SSP scenarios was used. It is reference pathways describing plausible alternative trends in the evolution of society and ecosystems over a century timescale, in the absence of climate change or climate policies".*

3. How are the mild, moderate, and extreme droughts defined? Which SPEI threshold are used?

**Authors' response:** *Thanks for your comments. Mild, moderate, and extreme droughts are graded by the value of SPEI. The categories and SPEI threshold were shown in Table 1 in Page 12. To make it more clear, the definition of each drought grade was supplemented in Table 1, and the statement "The categorization of drought grade by SPEI and its probability as well as the definition of each grade of drought" was added in Page 4 line 10 (Section 2.2).*

**Table 1. Drought grade categories and probability in the SPEI and its definition.**

| SPEI | Categories | Probability | Definition |
|---|---|---|---|
| >-0.5 | Normal and wetness | 69.15% | Precipitation is normal or more than normal, surface is wet and there is no drought |
| -1.0~-0.5 | Mild drought | 14.98% | Precipitation is less than normal, surface air is dry, and soil moisture is insufficient |
| -2.0~-1.0 | Moderate drought | 13.59% | Precipitation continued to be less than normal, surface is dry, soil moisture is insufficient, which has a certain impact on crops and ecological environment |
| ≤-2.0 | Extremely drought | 2.28% | Soil moisture is seriously deficient for a long time, which has a serious impact on crops, ecological environment, industrial production as well as drinking water for people and animals |

4. Isn't SPEI-12 related to hydrological drought? The authors implicitly suggest that the two are completely unrelated.

**Authors' response:** *Thanks for your suggestions. According to the definition of meteorological and hydrological drought. Meteorological drought is related to the precipitation deficit over a prolonged period of time. The commonly used meteorological drought indicators include SPI, PDSI and SPEI. Hydrological drought is generally related to the deficit of surface runoff, streamflow, reservoir, or groundwater. The commonly used hydrologic drought indicators include Palmer Hydrologic Drought Index (PHDI), runoff or streamflow percentile, Standardized*

*Runoff Index (SRI), or reservoir level. And the SPEI was used as hydrological drought indices in some research as well. Our inappropriate description led to a misunderstanding, so we have supplemented the description "SPEI is commonly applied as an indication of meteorological droughts and, to a lesser extent, hydrological droughts (Zargar et al., 2011; Hao et al., 2018)." in Page 3 line 25-26 (Section 2.2).*

[revised manuscript text omitted]

**Supplement**

[Figure]

Legend:
- Northeast region
- Southwest region
- Southeast coastal region
- Qinhai-Tibet region
- Inner Mongolia-Xinjiang region
- Shanxi-Shaanxi-Gansu-Ningxia region
- Lower reaches of the Yellow River region
- Middle and lower reaches of the Yangze River region

**Fig. S1 Regional divisions of China's population**

[Figure]

**Fig. S2 Spatial distribution of drought frequency for (a) the reference period (1986-2005) and (b) 1.5°C global warming scenario (2020-2039 in RCP2.6).** 1-3, mild droughts (1), moderate droughts (2), extreme droughts (3).

[Figure]

**Fig. S3 Spatial distribution of population and geo-demographic demarcation line (Hu line) of China in reference period (1986-2005).**

[Figure]

5  **Fig. S4 Change in population, in number (a) and percentage (b), between the reference period (1986-2005) and 1.5 °C global warming scenario (2020-2039 in RCP2.6).**

---

## Author Response (AR3)

Response to Editor's comments

Dear Editor:

   Thank you for your letter and the comments concerning our manuscript entitled "Population exposure to droughts in China under 1.5°C global warming target" (ID: esd-2017-100). Those comments are all valuable and very helpful for revising and improving our manuscript. We studied comments carefully and made corrections in the manuscript. The response to the comments are as follow:

1. In the abstract, please remove the reference to the Hu line as this is not widely known outside China. Perhaps rather describe this as the east vs. west of China or similar.

**Authors' response:** *Thanks for your suggestions. The statement was replaced to "Results revealed that population exposure to droughts on the east of China is higher than that on the west" in Page 1, line 17.*

2. The abstract misses a conclusion at the end.

**Authors' response:** *Thanks for your suggestions. The conclusion was supplemented at the end of abstract, Page 1, line 24-25. The statement is "The study suggested that reaching the 1.5°C target is a potential way for mitigating the impact of climate change on both drought hazard and population exposure".*

3. Page 2, line 4: droughts are "likely to increase" (instead of "increasing").

**Authors' response:** *Thanks for your advice. The statement was replaced to "likely to increase" in Page 2, line 5.*

4. Page 2, line 28: Reference for ISI-MIP Project?

**Authors' response:** *Thanks for your suggestions. Reference for ISI-MIP Project "(Warszawski et al., 2014)" was supplemented in Page 2, line 29.*

5. Page 2, line 29: Also, add a reference for the SSPs.

**Authors' response:** *Thanks for your advice. Reference for the SSPs "(O'Neill et al., 2014)" was added in Page 2, line 30.*

6. Page 2, last sentence: This evaluation... it does not provide effective strategies, but rather provides a basis for such strategies. Please adjust.

**Authors' response:** *Thanks for your suggestions. The statement was fixed to "This evaluation ... is expected to provide a basis for adaptation and mitigation strategies." in Page 2, line 33.*

7. Page 3, line 10: Define GMT.

**Authors' response:** *Thanks for your suggestions. GMT has been defined in Page 1 line 27. To make it clearer, we also added its definition "global mean temperature (GMT)" in Page 3, line 10-11.*

8. Page 3, line 20: add degree latitude/longitude to 0.5°.

**Authors' response:** *Thanks for your suggestions. We have added "degree latitude/longitude" to 0.5° in Page 3, line 21.*

9. Page 3, line 25: "normalized normalization". Please fix.

**Authors' response:** *Sorry for our inappropriate description. The statement was fixed to "and obtains the drought index value by standardizing" in Page 3, line 26.*

10. Page 4, line 14: "which is referred to Jones et al. -- What do you mean?

**Authors' response:** *Thanks for your comments. It means the definition of population exposure to droughts used in this study is referred to Jones' study about exposure to heat extremes. Sorry for our inappropriate writing. The statement was replaced to "the annual average percentage of mild, moderate, and extreme droughts multiplied by the number of people exposed to that outcome (Jones et al., 2015)." in Page 4, line 13-15.*

11. Page 8, line 21: What you list there are not four key conclusions, but is the summary of what you find. This is something different. The last sentence of the paragraph is more of a conclusion. Please adjust the text accordingly.

**Authors' response:** *Thanks for your suggestions. We have adjusted the text accordingly. Page 8, line 21: The statement was replaced to "This study leads to four key findings". Page 8, line 29: The statement was replaced to "To conclude, in the 1.5°C global warming scenario…".*

12. Figure 2: Define red line.

**Authors' response:** *Thanks for your suggestions. Definition of red line was supplemented in Page 13, line 3-5, the caption of Figure 2. The statement is "Red line is Hu line, an imaginary line that diagonally divides the area of China into two parts, stretching from the city of Heihe in Heilongjiang Province to Tengchong in Yunnan Province, which is also called the 'geo-demographic demarcation line'.".*

---

## Author Response (AR4)

Dear Editors and Reviewers:

Thank you for your letter and for the reviewer's comments concerning our manuscript entitled "Population exposure to droughts in China under 1.5° C global warming target" (ID: esd-2017-100). Those comments are all valuable and very helpful for revising and improving our manuscript. We studied comments carefully and made corrections in the manuscript. The texts marked in red, orange, green, blue and purple in the revised manuscript are modifications according to comments from reviewer #1, 2, 3, 4 and editor respectively. The response to comments are as follow:

**Response to reviewer #1**

1. Please define "risk" and "exposure" in introduction.

**Authors' response:** *Thanks for your suggestions. We have supplemented the definition of "risk" and "exposure" in P1 Line 31 (Section 1 Introduction). The statement is: "Risk is often represented as the probability of occurrence of hazardous events or trends multiplied by the impacts if these events or trends occur, it results from the interaction of hazard, exposure, and vulnerability (Field et al., 2014). Therefore, exposure assessment is one of the most important aspect of disaster risk assessment. Exposure usually refers to the presence of people, livelihoods, species or ecosystems, environmental functions, services, and resources, infrastructure, or economic, social, or cultural assets in places and settings that could be adversely affected (Field et al., 2014)."*

2. Rephrase P3 Line 12 to 13.

**Authors' response:** *Thanks for your advice. The statement was rephrased to "Combined the characteristics of the Standardized Precipitation Index (SPI) (McKee et al., 1993) at multiple scales and Palmer Drought Severity Index (PDSI) (Palmer, 1965) which is sensitive to warming, SPEI was proposed by Vicente-Serrano et al. (2010)" in Section 2.2.*

3. Please give more details on SPEI calculation.

**Authors' response:** *Thank you. We have added the statement "The SPEI reflects the*

*change in water deficit using the Log-logistic probability distribution function, and obtains the drought index value by normalized normalization." and "The radiation coefficient used is based on the radiation calibration results in China provided by Yin et al. (2008)". In Addition, detailed calculation process of potential evapotranspiration as well as procedure used to derive the SPEI and the set parameters were also supplemented in Section 2.2.*

4. Please define the "Hu line" and provide a brief introduction

**Authors' response:** *Thanks for your suggestions. We have supplemented the definition and a brief introduction of "Hu line" in Section 3.1. The statement is "(The variation in demographic change is clear when comparing the two sides of the Hu line), which is an imaginary line that diagonally divides the area of China into two parts, stretching from the city of Heihe in Heilongjiang Province to Tengchong in Yunnan Province. It is also called the "geo-demographic demarcation line"; the west of the line occupies 56.2 % of the area of China, but only 5.9 % of the population, while the east of the line occupies 43.8 % of the area, but 94.1 % of the population (Fig. S3)." In addition, we have added the Hu line in Fig 2、 Fig S3 and S4 so that the statement and figures would be easily understood.*

5. P5 Line 5, two scenarios?

**Authors' response:** *Sorry for our incorrect writing of "two scenarios". We have corrected the statement to "the reference period and the 1.5°C global warming scenario" in Section 3.2.*

6. Figure 5, figure (a) and figure (b) almost the same, so I suggest to add a total number of population.

**Authors' response:** *Thanks for your suggestions. Figure 5 shows the cumulative probability projected change drought frequency and population exposure in order to reflect the change of frequency and exposure of the three grades of droughts. Therefore, the change of population is not included. Figure (a) and figure (b) are similar because most of the probabilities of increase in frequency (a) and exposure (b) are near 50%, but there are some differences between the two figures. For example, the probability of*

*decrease of extreme droughts in frequency and exposure is 61.77% and 71.83 %*
*respectively. Besides, we have shown the change of population both in number and*
*percentage in Fig S4. Of course we also think the number of population is important,*
*the suggestion is valuable, so we added the spatial distribution of population of China*
*in reference period in Fig S3.*

7.  P7 "Results suggest that reaching the 1.5° C target is a potential mechanism for
mitigating the impact of climate change on droughts." It is not very clear.

**Authors' response:** *Thanks for your comments. The statement is based on the results*
*from our study. To make it more clear, we have rewritten this statement to "Fourth,*
*probabilities of increasing or decreasing total drought frequency are approximately*
*equal (49.86 % and 49.66% respectively), while the frequency of extreme drought is*
*likely to decrease (71.83 % probability) in 1.5°C global warming scenario. Results*
*suggest that in the 1.5°C global warming scenario, the contribution of climate change*
*is significantly less than demographic change and drought frequency will not increase*
*distinctly compared to reference period, which indicates that reaching the 1.5°C target*
*is a potential mechanism for mitigating the impact of climate change on both droughts*
*and population exposure." in Section 5.*

**Response to reviewer #2**

1. While I believe the study is very well conceived and the paper is very well written, I
have to object to one of the author's primary conclusions. I do not believe an increase
in exposure of 6.97 million persons constitutes a "substantial" increase. If anything I
would argue that it is quite the opposite. In 2030 6.97 million persons represent roughly
0.5% of the projected Chinese population under SSP1 (1.359 billion). In short, as
currently contextualized, the results/projections are a bit misleading as the increase in
exposure is rather unremarkable.

   I would suggest two possible pathways to remedy this issue. First, the authors might
reframe this result to highlight the importance of achieving the goals of the Paris Accord
within the context of Chinese droughts. This study finds that doing so will limit the
potential damage incurred by climate change. Second, this finding might be supported
by adding an additional scenario, such as an SSP2/RCP4.5, SSP3/RCP4.5, or
SSP5/RCP8.5 combination to illustrate the avoided impacts of achieving Paris. The

second suggestion entails significantly more work, and may be better thought of as future work, but at the very least I would like to see the paper reframed to better fit with the results.

**Authors' response:** *Thanks for your suggestions. The statement was rephrased in Section 4 and 5. The modifications are as follow:*

*P7 Line 17-19, We have added the statement "(The results indicated that average annual population exposure to droughts in the 1.5 ºC global warming scenario would increase by 6.97 million compared to the reference period,) roughly 0.51 % of the projected Chinese population under the SSP1 scenario in 2030. The increase in exposure is rather unremarkable, suggesting that achieving the 1.5 ºC target may limit the potential damage incurred by climate change."*

*P8 Line 3-5, We have added the statement "In future studies, we would like to evaluate population exposure for high GHG emission pathways, i.e., RCP4.5/SSP2 and RCP8.5/SSP3, and compare with the results from RCP2.6/SSP1 to illustrate the impacts of achieving the 1.5 ºC target."*

*P8 Line 24 We have revised the statement "a substantial increase" to "a slight increase".*

2. Page 3 lines 27-29: I would suggest rewriting as "The impact of population was calculated by holding climate constant, that is, the frequency of mild, moderate, and extreme droughts in the reference period multiplied by the population in the SSP1 scenario" (as opposed to ..."the population in the 1.5C global warming scenario). You want to convey to the reader that you are holding climate constant and allowing population to vary, so use the SSP as opposed to the temperature target.

**Authors' response:** *Thanks for your advice. The statement was rewritten to "The impact of population was calculated by holding climate constant, that is, the frequency of mild, moderate, and extreme droughts in the reference period multiplied by the population in the SSP1 scenario. Similarly, when calculating impact of climate, the population was held constant, that is, the frequency of mild, moderate, and extreme droughts in the RCP2.6 scenario was multiplied by the population in the reference*

*period" in Section 2.4.*

3. Page 4 line 4: I am assuming exposure is expressed in "average annual" population counts. I would suggest adding this terminology up front in Section 3.2 (e.g., "The average annual aggregate exposure.....)

**Authors' response:** *Thanks for your suggestions. We have added the express "average annual (aggregate exposure)" in Page 6 line 2 (Section 3.2) as well as Page 7 line 16 (Section 4).*

**Response to reviewer #3**

1. The standardized precipitation evapotranspiration index (SPEI) used in this study is an index of meteorological drought. Meteorological droughts do not necessarily coincide with agricultural, hydrological, or even socio-economic drought Thus, meteorological droughts have only limited direct relevance to people. In addition, the SPEI defines meteorological drought as departure from the mean climatic water balance (precipitation minus potential evapotranspiration) in multiples of standard deviations. For example, a value of -1 marks an event that deviates by one standard deviation from mean conditions. By definition, 15.9% of all time steps will be classified as -1 or less. It is obvious that such an indicator does not provide a measure of dryness in an absolute sense. Under wet conditions with low temporal variability, most SPEI droughts are still wet in an absolute sense; under dry conditions, many very dry events may not be classified as drought by the SPEI. Despite these shortcomings, I do believe that assessing population exposure to changes in meteorological droughts under climate change is a valid research question. But the limitations of the employed indicator (and drought type) must be highlighted and discussed to avoid misinterpretation of the results. This is clearly lacking in the paper, which instead tends to overstate the meaning of population exposure to meteorological droughts (e.g., page 2, lines 8-11).

**Authors' response:** *Thanks for your suggestions. We have added the express about limitations of the employed indicator (and drought type) in Section 4. The modifications are as follow:*

*P8 Line 8-11, we have added the statement "There are many kinds of droughts:*

*meteorological, agricultural, hydrological, and socioeconomic. In this study, based on simulated climate data, we assessed population exposure to meteorological droughts under the 1.5 °C global warming target using the SPEI; however, the results do not necessarily coincide with agricultural, hydrological, or socioeconomic droughts. Therefore, we would like to assess population exposure to different kinds of droughts to determine their impacts on populations."*

*P8 Line 14-19, we have added the statement "For instance, SPEI was chosen in this study because it combines the characteristics of SPI and PDSI; however, it is limited by providing a measure of dryness in a relative rather than absolute sense. Selecting different drought indexes may lead to differences in drought hazard and population exposure results. Therefore, future studies could evaluate different drought indexes based on more advanced and higher resolution GCMs and RCMs (regional climate models), determine importance of sources of uncertainty, and generate assessment results that are more accurate and reasonable."*

2. The basic concept of the SPEI is to transform a time series of the climatic water balance into a time series of normally distributed index values with a mean of 0 and a standard deviation of 1. For this transformation, a probability distribution function is fitted to the empirical distribution of climatic water balance values. The fitted distribution function is then used to map the climatic water balance values to SPEI values corresponding to the same quantile. Performing the transformation for present day and future time periods with independently fitted distribution functions, will yield two SPEI time series with the same statistical properties. Any attempt to identify a climate change signal will fail with this approach as the signal is lost in the transformation. Therefore, a single distribution function (preferably estimated from the reference period) must be used for the transformation of both the reference and future time series to be able to detect changes in the frequency of drought events. It is not clear whether this has been done correctly in this analysis as the method sections only provides a very vague description of the SPEI calculation. However, the results and how they are presented indicate that separate distribution functions have been fitted to the reference and the future time period.

**Authors' response:** *Thanks for your advice. We have added detailed statements of SPEI calculation in Section 2.2 and supplemented probability distribution of SPEI for*

*different drought grades in Table 1.*

3. On page 2 line 32 the authors explain that the climate data from the five available GCMs had been averaged prior to the analysis. Averaging time series is never a good idea. But in the case of GCM time series and with the aim to calculate SPEI it is simply wrong. The argument that "combining multiple models has been to shown superior to a single model" only holds true for long term averages and only for the comparison to observations. The SPEI analysis must be performed for each GCM individually. The results can then be averaged while properly accounting for GCM uncertainty.

**Authors' response:** *Thanks for your comments. Our inappropriate description led to a misunderstanding of the analysis. We have calculated SPEI for each GCM initially and averaged the results for drought frequency and population exposure analysis. So we have replaced the description to* "*In this study, we synthesized the results of the five GCMs based on the separately calculated SPEI for each GCM, as combining results of multiple models has been shown to be superior to a single model (Zhou and Yu, 2006).*" *in Page 3 line 6-8 (Section 2.1). Besides, we have added uncertainty discussion including GCM uncertainty in Page 8 line 12-19 (Section 4). The statement was rephrased to* "*In addition, there are some uncertainties in estimating population exposure under climate change. The main sources include GHG emission scenarios (Maurer, 2007), GCMs (Kirono et al., 2011), calculating potential evapotranspiration, population prediction, and selection of the drought index (Burke and Brown, 2008). For instance, SPEI was chosen in this study because it combines the characteristics of SPI and PDSI; however, it is limited by providing a measure of dryness in a relative rather than absolute sense. Selecting different drought indexes may lead to differences in drought hazard and population exposure results. Therefore, future studies could evaluate different drought indexes based on more advanced and higher resolution GCMs and RCMs (regional climate models), determine importance of sources of uncertainty, and generate assessment results that are more accurate and reasonable.*"

4. The paper defines population exposure to drought as "the frequency of mild, moderate, and extreme droughts multiplied by the number of people exposed to them" and reports it as number of people. I don't think this is appropriate. Let's assume a moderate drought is found to occur over 10 % of the time in a given grid cell. Then, according to the above definition, 10 % of the total population in that grid cell would

be counted as exposed to moderate drought. This is strange because intuitively one would expect that all people in that cell will experience moderate drought conditions over 10 % of the time. It is possible that it is only the unit (population numbers) that is puzzling here and that it could be fixed by including the temporal dimension. However, under no circumstance should the population exposure obtained for different drought severity classes be added (as done on multiple occasions in the paper).

**Authors' response:** *Thanks for your comments. There are different kinds of definition of population exposure to extreme climate events and disasters. For example, Smirnov et.al (2016) defined "populations' exposure to extreme drought as the total number of people, in the world or in a country, living in grid cells where SPEI < −2." While the definition of exposure we used is referred to Jones et al (2015), which defined population exposure to heat extremes as "the annual average number of days with a maximum temperature above 35 ℃ multiplied by the number of people exposed to that outcome." To state more clearly, we have change the description to "Our measure of population exposure is the number of people exposed to mild, moderate, and extreme droughts. That is, the annual average percentage of mild, moderate, and extreme droughts multiplied by the number of people exposed to that outcome, which is referred to Jones et al. (2015)" in Page 4 line 13-15 (Section 2.3).*

*As for calculation of population exposure of different severity classes, it is referred to studies of Smirnov et al. (2016) and Sun et al. (2017) as is mentioned in Section 1, which is also widely used in relevant studies. Smirnov et al. (2016) assessed population exposure to extreme droughts while the study did not account for mild and moderate droughts. Sun et al. (2017) analyzed population exposure to moderate, severe and extreme droughts under 1.5 ℃ and 2.0 ℃ global warming scenarios, while the study ignored the impact of demographic growth on population exposure change. In this study, calculation of population exposure of different severity classes make the results more accurate, and is useful for relative importance analysis. In addition, it is also important for vulnerability and risk assessment in further studies.*

**Authors' response:** *Thanks for your comments. Mild, moderate, and extreme droughts are graded by the value of SPEI. The categories and SPEI threshold were shown in*

*Table 1 in Page 12. To make it more clear, the definition of each drought grade was supplemented in Table 1, and the statement "The categorization of drought grade by SPEI and its probability as well as the definition of each grade of drought" was added in Page 4 line 11 (Section 2.2).*

**Table 1. Drought grade categories and probability in the SPEI and its definition.**

| SPEI | Categories | Probability | Definition |
|---|---|---|---|
| >-0.5 | Normal and wetness | 69.15% | Precipitation is normal or more than normal, surface is wet and there is no drought |
| -1.0~-0.5 | Mild drought | 14.98% | Precipitation is less than normal, surface air is dry, and soil moisture is insufficient |
| -2.0~-1.0 | Moderate drought | 13.59% | Precipitation continued to be less than normal, surface is dry, soil moisture is insufficient, which has a certain impact on crops and ecological environment |
| ≤-2.0 | Extremely drought | 2.28% | Soil moisture is seriously deficient for a long time, which has a serious impact on crops, ecological environment, industrial production as well as drinking water for people and animals |

4. Isn't SPEI-12 related to hydrological drought? The authors implicitly suggest that the two are completely unrelated.

**Authors' response:** *Thanks for your suggestions. According to the definition of meteorological and hydrological drought. Meteorological drought is related to the precipitation deficit over a prolonged period of time. The commonly used meteorological drought indicators include SPI, PDSI and SPEI. Hydrological drought is generally related to the deficit of surface runoff, streamflow, reservoir, or groundwater. The commonly used hydrologic drought indicators include Palmer Hydrologic Drought Index (PHDI), runoff or streamflow percentile, Standardized Runoff Index (SRI), or reservoir level. And the SPEI was used as hydrological drought indices in some research as well. Our inappropriate description led to a misunderstanding, so we have supplemented the description "SPEI is commonly applied as an indication of meteorological droughts and, to a lesser extent, hydrological droughts (Zargar et al., 2011; Hao et al., 2018)." in Page 3 line 26-27 (Section 2.2).*

[revised manuscript text omitted]